# Transcranial Direct Current Stimulation to Facilitate Lower Limb Recovery Following Stroke: Current Evidence and Future Directions

**DOI:** 10.3390/brainsci10050310

**Published:** 2020-05-21

**Authors:** Samuel Gowan, Brenton Hordacre

**Affiliations:** 1Interdisciplinary Neuroscience Program, Department of Biology, University of Wisconsin—La Crosse, La Crosse, WI 54601, USA; 2IIMPACT in Health, University of South Australia, Adelaide, SA 5001, Australia; brenton.hordacre@unisa.edu.au

**Keywords:** stroke, leg, lower limb, transcranial direct current stimulation, tdcs, brain stimulation, rehabilitation, recovery, neuroplasticity

## Abstract

Stroke remains a global leading cause of disability. Novel treatment approaches are required to alleviate impairment and promote greater functional recovery. One potential candidate is transcranial direct current stimulation (tDCS), which is thought to non-invasively promote neuroplasticity within the human cortex by transiently altering the resting membrane potential of cortical neurons. To date, much work involving tDCS has focused on upper limb recovery following stroke. However, lower limb rehabilitation is important for regaining mobility, balance, and independence and could equally benefit from tDCS. The purpose of this review is to discuss tDCS as a technique to modulate brain activity and promote recovery of lower limb function following stroke. Preliminary evidence from both healthy adults and stroke survivors indicates that tDCS is a promising intervention to support recovery of lower limb function. Studies provide some indication of both behavioral and physiological changes in brain activity following tDCS. However, much work still remains to be performed to demonstrate the clinical potential of this neuromodulatory intervention. Future studies should consider treatment targets based on individual lesion characteristics, stage of recovery (acute vs. chronic), and residual white matter integrity while accounting for known determinants and biomarkers of tDCS response.

## 1. Introduction

Stroke is the second leading cause of death and third leading cause of adult disability globally [1]. With advancement in acute medical care, more people now survive stroke, but frequently require extensive rehabilitative therapy to reduce impairment and improve quality of life. For those that survive stroke, the damaging effects not only impact the individual and their family, but there is also increased burden on health unit resources and community services as the person leaves hospital, potentially requiring assistance to live in the community. Novel treatments that can enable restoration and enhance potential for stroke recovery are desperately needed and will have significant value for many aspects of stroke care.

True recovery from stroke impairment is underpinned by neuroplasticity. Neuroplasticity describes the brain’s ability to change in structure or function in order to help restore behavior following neural damage. Mechanisms of neuroplasticity are available throughout life but appear enhanced during critical periods of learning [2]. Across several animal studies, it has been shown that there is a period of heightened neuroplasticity that appears to open within several days following stroke [2,3,4] and correlates with rapid recovery [5]. In humans, the timing and duration of a similar critical period of heightened neuroplasticity are not clear, but it likely emerges early after stroke. Understanding the characteristics of a potential critical period of heightened neuroplasticity in humans is important for optimizing stroke rehabilitation and is the subject of current trials [6]. However, the importance of neuroplasticity for stroke recovery in humans is unequivocal, with imaging and physiological studies providing extensive evidence of brain changes correlating with improved behavior [7,8,9,10,11,12,13]. 

Transcranial direct current stimulation (tDCS) is a promising, non-invasive, method to induce neuroplasticity within the cerebral cortex and augment stroke recovery. Importantly, tDCS has potential to bidirectionally and selectively alter corticospinal excitability for up to one hour after stimulation [14,15]. Animal models indicate that tDCS modulates resting membrane potential, with anodal stimulation leading to neuronal depolarization and cathodal stimulation leading to neuronal hyperpolarization over large cortical populations [16]. Stimulation-induced changes may be potentiated by changes in intracellular calcium concentrations. For example, anodal tDCS applied to the surface of the rat sensorimotor cortex led to a rise in the intracellular calcium concentrations [17]. Local increases in calcium can result in short- and long-term changes in synaptic function [18]. In humans, pharmacological studies have also provided indirect evidence to suggest that tDCS after effects are mediated by changes in synaptic plasticity through mechanisms that resemble long-term potentiation (LTP) and long-term depression-like effects [19]. Oral administration of the NMDA-receptor antagonist dextromethorphan was found to suppress the post-tDCS effects of both anodal and cathodal stimulation, suggesting that tDCS after effects involve NMDA receptors [19]. Importantly, modulation of cortical activity with tDCS changes human behavior [20]. For example, in randomized sham-controlled trials, anodal stimulation of the motor cortex (M1) in the lesioned hemisphere was found to improve upper limb outcomes in chronic [21,22,23] and subacute stroke survivors [24,25,26], with behavior changes underpinned by increased cortical activity within the M1 [27]. Although much work remains to be performed regarding optimal stimulation doses, cortical targets and electrode montages, these studies provide some indication that tDCS may be beneficial in stroke recovery.

While there is indication that tDCS has potential to improve stroke recovery of the upper limb [28], there are comparatively fewer studies that have investigated tDCS for lower limb recovery after stroke. Lower limb rehabilitation is especially important, as the simple act of regaining the ability to walk has subsequent effects on the ability to engage in activities of daily living [29,30]. Furthermore, those receiving therapy targeting mobility have been shown to have reduced levels of depression and anxiety [31], which are important determinants of stroke recovery [32,33,34]. Therefore, novel interventions capable of enhancing lower limb recovery might improve not only lower limb motor performance but could have added benefit for stroke rehabilitation in general. The purpose of this review is to discuss tDCS as a technique to modulate brain activity and promote recovery of walking following stroke. Within this review, we will outline current studies that have investigated tDCS to improve lower limb motor performance in both healthy adults and people with stroke. Additionally, we propose a best-practice model of experimental design for lower limb tDCS to guide future application for lower limb stroke recovery.

## 2. Is it Possible to Modify Lower Limb Motor Networks with Transcranial Direct Current Stimulation?

One approach to modify activity of the lower limb motor network with tDCS is to target the M1, similar to studies involving the upper limb. However, targeted application with tDCS is challenging as, compared with upper limb representations, the lower limb M1 representations are more medial and deeper within the interhemispheric fissure (Figure 1). This presents two notable difficulties. First, the ability of targeted stimulation to the lower limb M1 within one hemisphere (e.g., the lesioned hemisphere in stroke) is challenging, as tDCS electrodes can be relatively large compared to the size of cortical representations, resulting in current spread that may inadvertently lead to stimulation within the opposite hemisphere. Second, the depth of the lower limb M1 representations may present a challenge to current penetration and depth with traditional tDCS applications. However, there is evidence to indicate that it is possible to modulate activity of the lower limb M1 with tDCS. Computational modelling has revealed that traditional anodal tDCS electrode montages (anode overlying the lower limb M1 and cathode overlying the contralateral orbit; Figure 1) can lead to the expected cortical excitability enhancement in the target cortex [35]. Indeed, reducing the size of the anode (3.5 cm × 1 cm) was found to improve the specificity of the current delivered to the cortex, while positioning the return electrode (cathode) to a more lateral position (T7/8 on the 10–10 EEG system) further improved current specificity, leading to greater changes in cortical excitability [35]. Experimental evidence also suggests that tDCS targeting the lower limb M1 can modify excitability. Jeffrey and colleagues [36] utilized an anodal-tDCS montage (2 mA, 10 min) over the lower limb M1 and found that motor-evoked potentials (MEPs) of the tibialis anterior muscle increased by as much as 59% compared to sham conditions. Along similar lines, 10 sessions of anodal tDCS (2 mA, 10 min) targeting the lower limb M1 was found to increase the amplitude of MEPs recorded from the paretic tibialis anterior compared to sham stimulation [37]. This empirical evidence provides some support to the computational modelling to suggest that the use of tDCS targeting the lower limb M1 can modify corticospinal excitability. 

Although M1 has received attention as a stimulation target to modify excitability of the lower limb M1, there is potential for cerebellar tDCS to induce similar, or possibly more prominent, behavioral and neurophysiological changes. It is noteworthy that a computational modelling study that compared electrode montages targeting M1 and the cerebellum found that cerebellar stimulation produced substantially higher electric field strengths in the target area compared to M1 stimulation, suggesting the cerebellum may indeed be a suitable target for tDCS [38]. Behaviorally, the cerebellum contributes to motor planning, learning, and control; this influence is in part mediated by connections to M1 via the cerebellothalamocortical tracts, previously reported to play a key role in motor skill learning in mice [39]. Although this stimulation technique has received comparatively little attention compared to M1 stimulation, there is some indication that it is possible to modify cerebellar excitability in a focal and polarity specific manner [40]. Whether cerebellar tDCS is required to modify excitability of M1 for behavioral change is unclear. However, if a desired outcome was to modify M1 excitability with cerebellar stimulation, a pertinent challenge would be whether cerebellar tDCS can achieve the specificity required to precisely target the lower limb M1 in one hemisphere. Although speculative, one approach could be to pre-activate M1 through a contralateral lower limb motor task in order to bias the effects of tDCS towards those networks activated to perform the task. In support, there is some evidence in the upper limb that performance of a task during cerebellar tDCS does interact with the change in M1 excitability [41].

## 3. Transcranial Direct Current Stimulation to Improve Lower Limb Motor Performance in Healthy Adults and People with Stroke

In healthy adults, tDCS has proven beneficial for lower limb motor performance. For example, across two separate studies, anodal tDCS (2 mA, 10 min) applied to the lower limb M1 was found to transiently enhance the maximal leg pinch force [42] and ankle choice reaction time [43] compared to sham stimulation. Along similar lines, both anodal and cathodal cerebellar tDCS (1 mA, 15 min) were reported to improve ankle target-tracking accuracy [44], while cathodal cerebellar stimulation (1 mA, 9 min) was found to impair balance control in healthy adults [45]. Furthermore, a meta-analysis including 17 randomized controlled trials (629 healthy adults) demonstrated enhanced motor learning following anodal cerebellar tDCS in the short (<24 h) and long term (>24 h) [46]. Trials included within this meta-analysis used similar tDCS stimulation parameters to studies targeting M1 (1–2 mA, 15–20 min). However, despite evidence of some positive results, it is noteworthy that several studies have reported tDCS to have no impact on lower limb motor performance. In a triple blind, sham-controlled study, anodal tDCS (2 mA, 10 min) applied to M1 for seven sessions over 3 weeks was found to be ineffective at enhancing lower extremity strength [47]. Similarly, cathodal tDCS (2 mA, 20 min) applied to M1 was found to have no effect on lower limb tracking accuracy task, despite anodal tDCS (2 mA, 20 min) improving performance [44]. 

Similar to studies that have found positive effects in healthy adults, tDCS has proven beneficial for lower limb motor performance and learning in people with stroke. In a double-blind cross-over study, maximal knee extension force was significantly increased compared to sham in subcortical stroke survivors who received a single session of anodal tDCS to M1 for 10 min at 2 mA [48]. Improvements in motor performance persisted for 30 min following tDCS. Similarly, a single session of anodal tDCS (2 mA, 10 min) to M1 was found to enhance tracking error for an ankle task [49]. Using a slightly different montage known as dual-tDCS with the anode over the ipsilesional M1 and cathode over the contralesional cortex, a single session of tDCS (2 mA, 20 min) prior to conventional physical therapy was found to improve sit-to-stand performance [50]. However, similar to healthy adults, there appears to be some variability in response to tDCS. For example, robotic assisted gait training combined with tDCS (1.5 mA, 7 min) delivered for 10 sessions over two weeks was found to have no additional effect compared to robotic gait training with sham tDCS [51]. Along similar lines, anodal tDCS (2 mA, 10 min) delivered for 20 sessions over four weeks combined with robotic training had no additional benefit over sham tDCS at improving gait speed or gait quality [52]. 

Poor reliability of induced behavioral change following tDCS is not a challenge limited to lower limb studies. Rather, it is an issue facing the wider field of neuromodulation [53]. To address this issue, much work has been directed towards identifying determinants of response to non-invasive brain stimulation of the upper limb M1 representations [54,55,56]. These individual characteristics which are reported to influence response to tDCS provide some mechanistic insight to understand how current may influence brain activity and could server as potential biomarkers in future studies. Furthermore, much work is being conducted in both upper and lower limb M1 studies to identify optimal electrode montages, stimulation durations and intensities to improve response reliability to tDCS [35,57,58,59,60].

## 4. Principles of tDCS Application in Stroke

Approaches to apply tDCS targeting the lower limb in stroke are largely based on previous work performed in the upper limb. A dated, but commonly used, model to guide application of tDCS targeting the upper limb in stroke is the interhemispheric imbalance model (Figure 2). The model stipulates that after stroke, excitability of the ipsilesional hemisphere is suppressed, leading to reduced excitability of descending pathways and reduced interhemispheric inhibition from the ipsilesional to the contralesional hemisphere via transcallosal pathways. The result is a relative overall increase in excitability of the contralesional hemisphere leading to upregulation of descending pathways and increased inhibition from contralesional to ipsilesional hemisphere, further suppressing activity from the ipsilesional hemisphere [11,61,62,63,64,65,66]. This imbalance in excitability between ipsilesional and contralesional hemisphere has been associated with post-stroke upper limb impairment [65]. As a result, many tDCS studies have attempted to balance excitability between hemispheres following stroke by either applying anodal tDCS to the ipsilesional hemisphere to increase excitability and/or cathodal tDCS to the contralesional hemisphere to suppress excitability [20,21,23,24,26,67,68]. 

However, despite the popularity of this model, more recent evidence suggests that this model is either oversimplified or incorrect all together [69]. The authors propose a new model, known as the bimodal balance-recovery model, which depends on the severity of the stroke and structural reserve of white matter pathways. For minor strokes where residual integrity of white matter pathways is maintained, or there is high structural reserve, the interhemispheric imbalance model will dominate. Therefore, restoring balance in excitability between hemispheres is likely to be behaviorally beneficial. However, in more severe stroke where integrity of critical white matter pathways, such as the corticospinal tract, is compromised, or in the case of low structural reserve, a vicariation model dominates. In the vicariation model, activity within residual networks substitute for lost function through neuroplastic processes. Severe stroke is often accompanied by increased levels of neural activity within the unaffected hemisphere, which is likely a compensatory response for extensive neural damage in the lesioned hemisphere [70]. Determining which model is dominant is important, as the two models can lead to opposing predictions about the optimal treatment strategy with tDCS. Specifically, interhemispheric imbalance model would suggest increasing excitability of the lesioned hemisphere and/or decreasing excitability of the non-lesioned hemisphere should lead to behavioral improvements. Conversely, the vicariation model would tend to support increasing excitability within residual networks, such as the non-lesioned hemisphere, would promote recovery. An example of the variation in response to tDCS between these two models was demonstrated with cathodal tDCS (1 mA, 20 min) applied to the non-lesioned M1 in a sample of subcortical stroke survivors with varying upper limb impairments [71]. Those with mild impairment and residual integrity of the corticospinal tract benefited from cathodal tDCS, while stimulation for those with severe stroke and compromised integrity of the corticospinal tract led to worse behavioral outcomes. However, it is worth noting that this study investigated upper limb outcomes and the role of the corticospinal tract may differ for the lower limb. Although there is some indication that excitability of M1 and the corticospinal tract is associated with lower limb activity [72,73], there are likely significant differences in upper and lower limb motor control. Despite this, it is highly reasonable to assume that structural reserve and stroke severity are likely to be important characteristics that might influence response to tDCS for the lower limb. We therefore suggest that future studies seek to further understand the role of structural integrity of white matter pathways in guiding application of tDCS for lower limb stroke recovery.

## 5. Quantifying Response to tDCS Application

Understanding the induced effect of tDCS on the motor system is critical to improve reliability and identify optimal techniques to modify human behavior. We propose that future studies should quantify both behavioral and neurophysiological outcomes when performing tDCS experimental studies to modify lower limb activity. However, there are currently numerous behavioral performance tests utilized across studies, making it a debilitating challenge for meta-analyses to combine outcomes from several studies. It is noteworthy that several previous lower limb tDCS studies have used highly sensitive behavioral assessments such as ankle tracking error [44,49], ankle choice reaction time [43] or strength assessments [42,47,48] in both healthy adults and stroke survivors. The sensitivity of these outcome measures may help to identify tDCS induced changes and are likely to be appropriate outcome measures particularly for healthy adults that do not have an underlying impairment in lower limb behavior. However, very few studies have investigated tDCS induced changes in impairment or activity-based measures directly related to gait or balance. Early evidence suggests that tDCS may have capacity to modify gait and balance outcomes in people with stroke [50,52], with these improvements likely to be more important for restoring mobility capacity and activities of daily living following stroke than highly sensitive assessments such as tracking error on an ankle tracing task. Furthermore, the use of common clinical outcome assessments for gait and balance might facilitate comparison of tDCS as an intervention with other treatments for mobility or balance in people with stroke. Moreover, the recent Stroke Recovery and Rehabilitation Roundtable produced an international consensus statement on standardized outcome measures in stroke trials [74]. For the lower limb, the 10 m walk test and Fugl-Meyer Lower Extremity were identified as a core outcome at all stages of stroke recovery (acute, subacute, chronic). For mobility, it was also identified that the measures left-right symmetry of spatial-temporal gait parameters and gait kinetics are also likely to be important outcome measures.

Neurophysiological measures can provide insight to physiological changes induced by tDCS and should be viewed as complementary to behavioral measures to help determine neural mechanisms that might enable improved lower limb behavior following tDCS. Commonly, neurophysiological changes induced by tDCS are quantified using transcranial magnetic stimulation (TMS) or neuroimaging (Figure 3). Single-pulse TMS with electromyogram (EMG) electrodes positioned overlying the muscle of interest can quantify the change in excitability of the descending pathways that innervate lower limb muscles following tDCS. Common muscles selected to record corticospinal excitability of the lower limb include the tibialis anterior [36,37], as it is a relatively large muscle, has a major impact on walking and gait, and is superficial and easy to palpate for EMG application. Alternatively, corticospinal excitability of the quadriceps femoris has been shown to correlate with gait performance in other clinical populations [75]. A challenge for recording MEPs with TMS is the depth of cortical representations within M1. It may be possible to overcome these challenges using deeper penetrating TMS coils, such as a double cone coil, for higher stimulation intensities or pre-activating the target muscle [75,76,77]. These methodological techniques are not without limitations, such as current spread from higher intensity stimulation or the influence of pre-stimulus muscle activity on corticospinal excitability. However, in well-designed experimental paradigms, they may be appropriate strategies to understand the physiology of the lower limb M1. Alternatively, neuroimaging may be an appropriate technique to quantify changes in neural activity of lower limb cortical representations following tDCS. For example, using functional magnetic resonance imaging in a sham-controlled study testing healthy adults, anodal tDCS (2 mA, 15 min) applied to the lower limb M1 led to increased blood oxygen level-dependent signal in multiple brain areas associated with leg performance compared to sham stimulation [78]. Although comparatively fewer studies have used neuroimaging techniques to quantify brain changes induced with tDCS, the increased spatial resolution and ability to quantify neural changes from deeper structures, such as the lower limb M1, might suggest that this is an appropriate outcome measure. Where possible, it may be advantageous to utilize a combination of behavioral, physiological, and neuroimaging techniques to quantify the degree of change induced by tDCS.

## 6. Biomarkers and Determinants of Response to tDCS

Without doubt, there are several challenges for the application of tDCS to improve lower limb behavior following stroke. First, as previously mentioned, tDCS responses are known to be variable. A large body of work has identified several determinants of response to brain stimulation in the upper limb that are equally likely to influence lower limb brain stimulation. Briefly, these determinants include age, gender, history of synaptic activity, genetics, pharmacology, neurophysiological characteristics of the stimulated cortex and functional connectivity of the target network [54,55,56]. Furthermore, the application of tDCS in stroke is likely to induce additional variability sources. Stroke is a heterogeneous condition and there is evidence that characteristics of the lesion influence how brain stimulation exerts its influence. For example, repetitive TMS applied to the lesioned M1 increased excitability and improved upper limb behavior for people with subcortical stroke, but was ineffective for people with cortical stroke [79]. In a larger study of 60 subacute stroke survivors, it was reported that anatomical lesion location was an important determinant of response to repeated sessions of brain stimulation. Patients with lesions that involve the cortex did not appear to respond to stimulation and demonstrated reduced improvement in motor activity compared to those with subcortical lesions only [80]. Furthermore, it has been shown that the hemodynamic responses to brain stimulation also appear different between cortical and subcortical lesions with velocity of blood flow significantly increased after stimulation in subcortical stroke, but was less prominent in cortical stroke [81]. Different stages of stroke recovery also appear to influence neuroplasticity processes, with a spontaneous upregulation in neuroplasticity emerging early after stroke [2,3,5]. Therefore, the application of tDCS might have differing influences depending on whether stroke participants are acute or chronic. Finally, in the upper limb, there is good evidence that integrity of descending white matter pathways is an important determinant of potential for recovery from therapeutic interventions in people with stroke [82]. While there are likely differences in the role of descending pathways for upper and lower limb motor control, it is not unreasonable to assume that motor pathway integrity would possibly have some influence over response to rehabilitative therapies. Together, these characteristics of tDCS and response to stimulation in people with stroke provide some indication to the mechanisms of action of brain stimulation in the presence of a lesion. We suggest that future studies involving tDCS should report characteristics which are known to influence response to stimulation, such as age, gender and pharmacology [54,55,56]. Where future studies are investigating people with stroke, we also suggest that key clinical characteristics of stroke location, stage of recovery (chronic vs acute) and motor pathway integrity also be determined and clearly stated. Lastly, given the inherent challenges of response variability with tDCS, and in particular in people with stroke, we suggest that all experimental designs should include an appropriate sham-controlled condition. This would help ensure that any observed responses are indeed the effect of tDCS and not induced by any of the several determinants of response variability.

## 7. Limitations

As with all narrative reviews, readers should be aware that the conclusions are based on an unsystematic, and therefore a potentially distorted, review of the literature. Although every effort was made to provide a balanced review of the literature, it is possible that a systematic approach would provide a more comprehensive and accurate summary of tDCS to facilitate lower limb recovery in people with stroke.

## 8. Conclusions

Transcranial direct current stimulation is a novel technique that might be capable of inducing a neuroplastic response to improve lower limb motor performance following stroke. While early studies provide some indication of both behavioral and physiological changes in brain activity following tDCS, much work still remains to be performed to demonstrate clinical potential. In order to thoroughly investigate the potentially beneficial role of tDCS to support lower limb recovery in stroke, we suggest that future studies strongly consider treatment strategies or stimulation targets based on individual lesion characteristics and residual white matter integrity. This might include the addition of current density modelling to identify optimal electrode montages to account for these unique neuroanatomical considerations in the post-stroke brain. Furthermore, to help overcome some of the inherent variability of tDCS, investigators should continue to explore, and where possible, report on known determinants of response variability for study participants. Finally, to facilitate cross-study comparison, we propose that future studies select behavioral outcome measures that are clinically relevant and comparable across different stroke treatments. Transcranial direct current stimulation is an intervention with significant promise, but future research must be considered and purposeful to ensure the full potential of this brain-modifying intervention can be evaluated.

## Figures and Tables

**Figure 1 brainsci-10-00310-f001:**
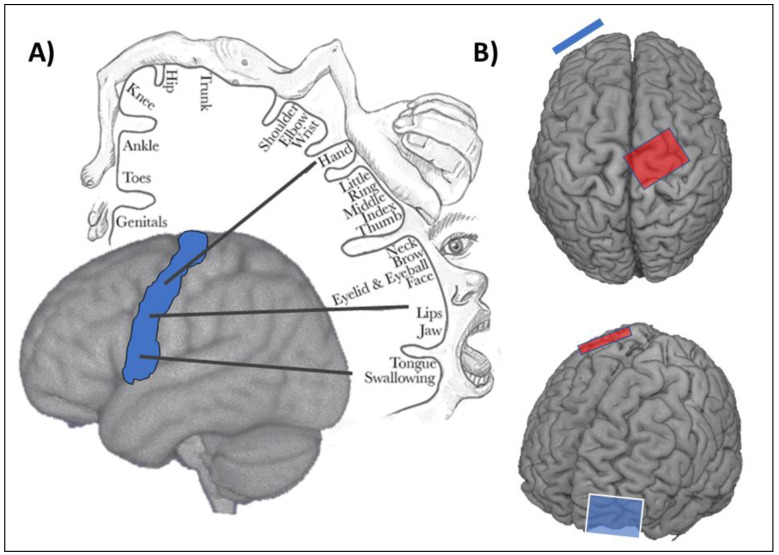
Transcranial direct current stimulation targeting the lower limb motor cortex in humans. (**A**) An example of the motor homunculus in humans. The motor strip on the cortex is highlighted in blue. Note that the lower limb representation is medial and deep within the motor cortex, presenting a challenge to target brain stimulation to this region. (**B**) An example of a standard transcranial direct current stimulation montage for targeting the lower limb motor cortex. The anode is shown in red and approximately overlies the lower limb motor cortex. The cathode (return) electrode is shown in blue and is typically positioned over the contralateral orbital region.

**Figure 2 brainsci-10-00310-f002:**
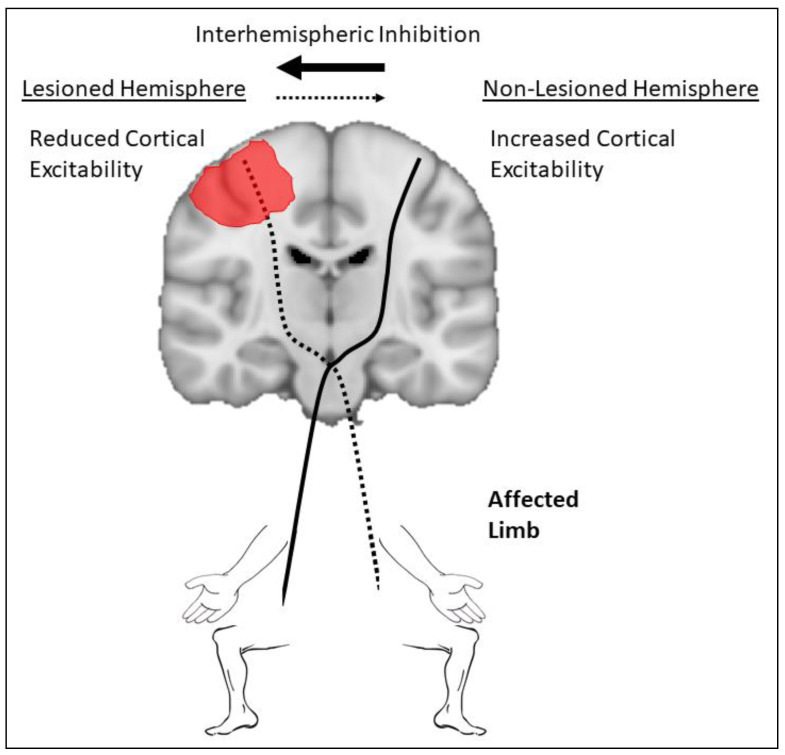
The interhemispheric imbalance model is commonly used to apply transcranial direct current stimulation. The lesion is shown in red within the cortex. Note that excitability of the lesioned hemisphere is reduced, leading to a decrease in excitability of descending and interhemispheric pathways (shown as a dotted line). Interhemispheric inhibition is imbalanced between hemispheres, shown as a dotted line for reduced interhemispheric inhibition at the top of the image, and a thick solid line for increased interhemispheric inhibition. This imbalance in interhemispheric inhibition further suppresses excitability of the lesioned hemisphere.

**Figure 3 brainsci-10-00310-f003:**
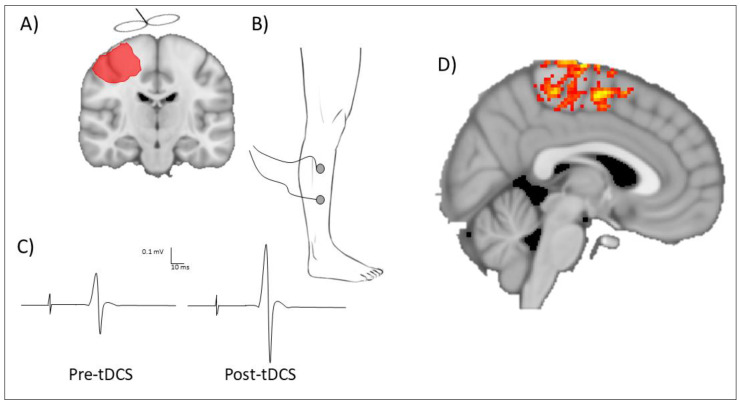
Techniques to quantify neurophysiological changes after lower limb transcranial direct current stimulation. One approach is to use transcranial direct current stimulation (tDCS) applied to the lower limb motor cortex (**A**) with surface electromyography recording motor-evoked potentials from a lower limb muscle on the paretic limb (for example, the tibialis anterior (**B**)). An example motor-evoked potential can be seen in (**C**) which depicts the amplitude of the motor-evoked potential increasing after transcranial direct current stimulation. The magnitude of change in motor-evoked potential amplitude is a marker of corticospinal excitability and can be used to quantify tDCS response. An alternative technique is to use neuroimaging approaches such as functional magnetic resonance imaging. (**D**) An example of neural activity within the lower limb motor cortex. Changes in neural activation can be compared as a marker of physiological changes in the brain.

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
