# Peer review of "Transcranial Direct Current Stimulation to Facilitate Lower Limb Recovery Following Stroke: Current Evidence and Future Directions"

_brainsci, 2020, doi:10.3390/brainsci10050310_

Round 1

Reviewer 1 Report

The purpose of work of Gowan and Hordacre is to discuss tDCS as a technique to modulate brain activity and promote recovery of walking following stroke. Additionally, the authors propose a best-practice model of experimental design for lower limb tDCS to guide future application for lower limb stroke recovery. This review is well written and potentially interesting for researchers who deal with the topic and for those who care and rehabilitate the stroke patient.

Aside from the small notes I have listed below, I think the main limitation of this work is that it is a narrative revision of the literature. So I ask the authors to add a sentence to the discussion that makes the reader aware that the conclusions drawn are based on an unsystematic, therefore potentially distorted, review of the literature.

Minor points:

Line 62, "in-direct" --> indirect

Line 162, In my opinion, the phrase "Similar to studies performed in healthy adults" is misleading because it suggests that all studies on healthy subjects found an effect, while the previous paragraph ends by citing several studies that have reported tDCS to have no impact on lower limb motor
157 performance. Therefore, I would change it to "Similar to studies that have found positive effects in healthy adults".

Reviewer 2 Report

The authors present a review paper on transcranial direct current stimulation (tDCS) to improve leg motor function after stroke. The paper first addresses general aspects of neuroplasticity. The study situation for direct transcranial stimulation leading to an improvement of leg motor function is discussed. The authors present the different principles of stimulation and methods to quantify the effect of direct current stimulation. Finally, biomarkers and determinants of this form of therapy are discussed. The authors conclude giving the therapeutic potential of this method and suggestions for further studies.

The manuscript is well written. It is reader-friendly and understandable. The three illustrations 1) on the principle, 2) on interhemispheric imbalance and 3) on the technique to quantify neurophysiological changes are clear and instructive. The bibliography is comprehensive and includes all the important works, so that the interested reader can further deepen the individual aspects of the topic. Overall, this is a well thought-out and carefully prepared overview.
